# Impact of a High-Fat Diet on the Gut Microbiome: A Comprehensive Study of Microbial and Metabolite Shifts During Obesity

**DOI:** 10.3390/cells14060463

**Published:** 2025-03-20

**Authors:** Md Abdullah Al Mamun, Ahmed Rakib, Mousumi Mandal, Udai P. Singh

**Affiliations:** Department of Pharmaceutical Sciences, College of Pharmacy, The University of Tennessee Health Science Center, 881 Madison Avenue, Memphis, TN 38163, USA; mmamun3@uthsc.edu (M.A.A.M.); arakib@uthsc.edu (A.R.); mmandal1@uthsc.edu (M.M.)

**Keywords:** obesity, gut microbiota, dysbiosis, metabolites, metabolic profile

## Abstract

Over the last few decades, the prevalence of metabolic diseases such as obesity, diabetes, non-alcoholic fatty liver disease, hypertension, and hyperuricemia has surged, primarily due to high-fat diet (HFD). The pathologies of these metabolic diseases show disease-specific alterations in the composition and function of their gut microbiome. How HFD alters the microbiome and its metabolite to mediate adipose tissue (AT) inflammation and obesity is not well known. Thus, this study aimed to identify the changes in the gut microbiome and metabolomic signatures induced by an HFD to alter obesity. To explore the changes in the gut microbiota and metabolites, 16S rRNA gene amplicon sequencing and metabolomic analyses were performed after HFD and normal diet (ND) feeding. We noticed that, at taxonomic levels, the number of operational taxonomic units (OTUs), along with the Chao and Shannon indexes, significantly shifted in HFD-fed mice compared to those fed a ND. Similarly, at the phylum level, an increase in *Firmicutes* and a decrease in *Bacteroidetes* were noticed in HFD-fed mice. At the genus level, an increase in *Lactobacillus* and *Ruminococcus* was observed, while *Allobaculum*, *Clostridium*, and *Akkermansia* were markedly reduced in the HFD group. Many bacteria from the *Ruminococcus* genus impair bile acid metabolism and restrict weight loss. *Firmicutes* are efficient in breaking down complex carbohydrates into short-chain fatty acids (SCFAs) and other metabolites, whereas *Bacteroidetes* are involved in a more balanced or efficient energy extraction. Thus, an increase in *Firmicutes* over *Bacteroidetes* enhances the absorption of more calories from food, which may contribute to obesity. Taken together, the altered gut microbiota and metabolites trigger AT inflammation, which contributes to metabolic dysregulation and disease progression. Thus, this study highlights the potential of the gut microbiome in the development of therapeutic strategies for obesity and related metabolic disorders.

## 1. Introduction

Over the past century, clinical microbiology has focused primarily on studying pathogens, but the significance of the commensal microorganisms that inhabit different body regions is becoming increasingly recognized [1,2]. The gut microbiome comprises living microorganisms often addressed as the “second human genome”. The gut microbiome comprises living microorganisms often addressed as the "second human genome" because of its potential role in navigating human health and influencing the body’s energy equilibrium [3]. Any dysregulation in energy intake and expenditure can be a major cause of obesity and related comorbidities [4]. Obesity is a growing health concern and requires attention due to its pandemic status in the modern world. Obesity is shaped by several factors, including genetics, eating habits, levels of physical activity, and gut microbiota, which all influence host metabolism as well as obesity [5,6].

Evidence has demonstrated that the transfer of gut microbiota from obese mice to germ-free mice on a standard diet induces obesity [7]. Nonetheless, the quantitative contribution of the gut microbiota to the host’s energy balance remains elusive. An imbalance in intestinal microbes can disrupt the function of gut barriers and gut-associated lymphoid tissues, allowing bacterial components to cross the intestinal wall to trigger the production of inflammatory cytokines and lead to insulin resistance and several metabolic diseases. Typically, the microbiome’s composition in obese individuals differs significantly from that in non-obese people, as shown in the genera *Akkermansia*, *Faecalibacterium*, *Oscillibacter*, and *Alistipes*, which are reduced in obese individuals [8,9]. These bacteria play crucial roles in maintaining gut health, gut barrier function, and producing anti-inflammatory mediators. On the contrary, HFDs often promote the growth of potentially harmful bacteria and fungi, including species from the genera *Dorea* and *Candida*, respectively [10]. The overgrowth of these bacteria can compromise the integrity of gut barrier function, allowing harmful bacterial components to trigger systemic inflammation and contribute to metabolic disorders such as insulin resistance and obesity [11].

Different lifestyles influence the microbiome by altering homeostasis and various diets, distinctly shaping the structure of gut microbiota [12]. HFDs can assist the growth of a mucus-degrading microbiome, whereas a fiber-rich diet reduces the prevalence of mucus-degrading microbes and promotes the growth of fiber-degrading SCFA-producing bacteria, thereby supporting mucosal barrier functions [13,14]. It has been shown that gut microbiota regulates energy and microorganisms’ ability to ferment dietary polysaccharides [15]. The production of SCFAs by the gut microbiota is heavily dependent on the composition and type of dietary fibers present in the diet [16]. Further, SCFAs have been shown to be involved in lipogenesis [17]. Additionally, SCFAs can attenuate the fasting-induced adipocyte factor (FIAF), which limits lipoprotein lipase (LPL), leading to triglyceride accumulation in host adipocytes [7]. Gut microbiota may also play a role in metabolic disturbances seen in obesity by initiating systemic inflammation [18] and LPS from gut microbiota and binding to toll-like receptors (TLRs), mainly TLR4 [19]. TLRs are well-known immune transmembrane proteins that can upregulate inflammatory cytokines and chemokines [20]. It has been shown that alterations in *Akkermansia muciniphila* induce the genetic expression of gut mucosal tight-junction proteins [21].

Recent advancements in next-generation sequencing (NGS) technologies have facilitated the detailed analysis of diverse gut microbiota, addressing the gap in microbiota studies traditionally reliant on culture-based methods. The Human Microbiome Project, led by the National Institutes of Health, identified over 70 bacterial phyla, with the majority belonging to *Actinobacteria*, *Bacteroidetes*, *Firmicutes*, and *Proteobacteria* [22]. Studies in the past suggested that imbalance in *Bacteroidetes* and *Firmicutes* in the intestinal tract of mice causes the ineffective absorption of carbohydrates and ultimately results in obesity [23]. Despite a lot of evidence, there are still gaps in our understanding of the role of gut microbiota in the development of obesity and how it mediates AT inflammation. In this study, we examined how a normal diet (ND) and an HFD modulate the gut microbiota populations which in turn alter AT inflammation and obesity. Using the 16S rRNA gene sequencing technique, we assessed the changes in gut microbiota in response to both HFD and ND in relation to the development and progression of obesity.

## 2. Materials and Methods

### 2.1. Animal and Diet Formulas

We purchased 6-to-8-week-old wild-type mice from Jackson Laboratories (Bar Harbor, ME, USA). The mice were housed under temperature-controlled conventional conditions for one week at the University of Tennessee Health Science Center (UTHSC)’s animal facility for acclimatization under a normal light/dark cycle before starting the experiment. This study was performed under a protocol (23-0451) from the UTHSC Institutional Animal Care and Use Committee (IACUC). The handling of mice and experimental methods were performed in such a way as to minimize pain and discomfort. Based on similar weights, mice were randomly divided into two experimental groups of 5 mice each (*n* = 5). One group of mice received a 10% Kcal normal diet (ND) or a 60% kcal high-fat diet (HFD) (Research Diets, New Brunswick, NJ, USA) for 8 weeks. The details of the diets are described below.

The HFD contained 26.2% proteins, 26.3% carbohydrates, and 34.9% fats (Research Diets, D12492). The ingredients were the following: casein, 80 mesh (200 g); L-Cystine (3 g); corn starch (0 g); maltodextrin 10 (125 g); sucrose (68.8 g); cellulose (Solka FLOC BW200, 50 g); soybean oil (25 g); lard (245 g); mineral mix, S10026 (10 g); dicalcium phosphate (13 g); calcium carbonate (5.5 g); potassium citrate 1 H_2_O (16.5 g); vitamin mix (V10001, 10 g); choline bitartrate (2 g); and FD&C Blue Dye #1 (0.05 g).

The ND contained 19.2% proteins, 67.3% carbohydrates, and 4.3% fats (Research Diets, D12450J). The ingredients were the following: casein, 30 mesh (200 g); L-Cystine (3 g); corn starch (506.2 g); maltodextrin 10 (125 g); sucrose (68.8 g); cellulose (BW200, 50 g); soybean oil (25 g); lard (20 g); mineral mix, S10026 (10 g); dicalcium phosphate (13 g); calcium carbonate (5.5 g); potassium citrate 1 H_2_O (16.5 g); vitamin mix (V10001, 10 g); choline bitartrate (2 g); FD&C Yellow Dye #5 (0.04 g); and FD&C Blue Dye #1 (0.01 g).

### 2.2. Fecal Sample Collection and DNA Isolation

Mice fecal samples were collected in 2 mL Eppendorf tubes after 8 weeks of ND and HFD feeding from all mice individually. During the collection of feces, all the animals were kept in a single cage to avoid intragroup cross-contamination. The sample was kept on ice and then stored at −80 °C until further analysis. A frozen aliquot (200 mg) of each fecal sample was used for extracting genomic DNA. According to the manufacturer’s instructions, DNA was extracted using a Qiagen QIAamp DNA Stool Mini Kit (Qiagen, Valencia, CA, USA, Cat. No.: 51604). DNA concentration and quality were measured using a nanodrop instrument (Thermo Scientific, Waltham, MA, USA). Extracted DNA was stored at −80 °C until further analysis. The DNA samples were sent to Novogene by our collaborator (Dr. Joseph F. Pierre, Ph.D., University of Wisconsin, Madison, WI, USA) for RNA library preparation and transcriptome sequencing on the HiSeq 2500 platform (Illumina, San Diego, CA, USA).

### 2.3. Bacterial Amplicon Sequencing and Statistical Analysis

We sent our sample to the Argonne National Laboratory (Chicago, IL, USA) for sequencing using an Illumina MiSeq DNA platform and a 150-nucleotide paired-end read length. The data were analyzed at UTHSC by the program Quantitative Insights into Microbial Ecology (QIIME), as described in [24]. The changes in bacterial structure were assessed through 16S rRNA V4–V5 amplicon sequencing, which enclosed a 12 bp barcode and Illumina 3′ adaptor sequences. The operational taxonomic units (OTUs) at 96% sequence similarities using open-reference OTU picking against the GreenGenes database (v13.8) were used for analysis. We further used all statistical analyses using the Calypso software (v8.84), as described in [25], with a minimum of 5000 sequences per sample. Some used reagents were also surveyed as sequencing controls, which failed to produce any amplification. The Shannon and Simpson indexes were used to analyze the alpha (α)-diversity changes in the microbiome. We also used beta (β)-diversity analysis via the Bray–Curtis method, as appearing as principal coordinate analysis (PCoA). Further, we used an unbiased analysis to generate a heatmap of the 100 most abundant microbial taxa, as calculated by Spearman’s rank correlation coefficients. The linear discriminant analysis (LDA)’s effect size (LEfSe) was calculated for significantly enriched taxa in both groups. We used Phylogenetic Investigation of Communities by Reconstruction of Unobserved States (PICRUSt) and predicted the metabolite pathways calculated by Spearman’s rank correlation coefficient, presented on a heatmap of the 100 most abundant pathways. Regression analyses of key operational taxonomic units (OTUs) with significant correlation were performed with multivariate analysis by linear models (MaAsLins) with 0.05 false discovery rate significance thresholding and 0.0001 minimum feature relative abundance filtering, similarly to [26]. In all experimental analyses, we used *t*-tests and ANOVA for all the comparisons in all bar charts at all levels of analysis.

## 3. Results

### 3.1. HFD Dysregulates the Composition and Decreases the Richness of the Gut Microbiota

A healthy microbiota is rich and varied, but factors like diet, stress, and pollutants can alter this imbalance. We investigated the effect of an HFD on the composition of the gut microbiota in mice by conducting 16S rDNA sequencing on mice fecal samples. We noticed that HFD consumption slightly decreased the bacterial richness compared to ND (Choa1; Figure 1A). However, the HFD-fed mice showed a slightly higher microbial diversity compared to those fed an ND, as indicated in the Shannon index (Figure 1B). Next, we analyzed β-diversity across two experimental groups, and principal component analysis (PCA) revealed a clear separation between the gut microbiota of mice fed an ND and those fed an HFD. On the principal coordinate analysis (PCoA) axis, both groups were situated in separate areas, indicating a change in the overall structure of the gut microbiota following HFD feeding. Bray–Curtis dissimilarity calculated the dissimilarity between two samples based on the counts of different species or features present in each sample. We noticed a significant difference in the gut microbiota composition according to the Bray–Curtis model (Figure 1C), as well as differences in the microbiome population in the samples in the same group in the case of the HFD and ND groups (Figure 1D). However, HFD feeding did not cause a notable change in the taxonomic diversity, which was analyzed by an unweighted paired-group method with an arithmetic mean (UPGMA) (Figure 1E). Taken together, our data indicated that the HFD modulated richness, α-, and β-diversity compared to the ND.

### 3.2. Modulation of Microbial Taxonomic Profiling in HFD-Fed Mice

The predominant components of the gut microbiota at the phylum level are *Firmicutes*, *Bacteroidetes*, *Actinobacteria*, and *Proteobacteria*. HFD consumption has been shown to induce obesity, which leads to the development of chronic disease, resulting from differential alteration in the microbiota, specifically a decrease in *Bacteroidetes* and an increase in *Firmicutes* [27]. Obesity studies have documented an increased abundance of *Firmicutes* concomitant with a reduction in *Bacteroidetes* [28]. In contrast to these findings, several studies reported no changes in this parameter and even documented a decreased *Firmicutes*/*Bacteroidetes* ratio in obese animals and humans [29,30,31]. Specifically, in most studies, individuals exhibiting lower bacterial diversity compared to lean individuals imply the presence of other compositional changes at the family, genus, or species level that may hold greater significance than the *Firmicutes*/*Bacteroidetes* ratio [32]. The previously reported discrepancies could be attributed to variations in sample processing and data analysis, such as differences in DNA extraction methods, the choice of primers for the amplified 16S rRNA region, sequencing techniques, and bioinformatic approaches, including the taxonomy database and assignment algorithms used [33,34]. In addition, the sample storage and DNA isolation method can influence the microbiota profile between studies [33,35]. The elevated *Firmicutes*/*Bacteroidetes* ratio did not correlate with the production of SCFAs observed in obese individuals, and *Bacteroidetes* predominantly produced acetate and propionate [36]. In the present study, the abundance of *Firmicutes* and *Proteobacteria* increased significantly, while the abundance of *Bacteroidetes* decreased in the HFD group compared to the ND group (Figure 2A). We also noticed a change in the diversity of major gut microbiota at the genus level (Figure 2B,C). At the genus level, abundance was measured with the total sum scaling (TSS) method, and it clearly showed a significant decrease in *Allobaculum* and *Akkermansia* in the HFD group compared to the ND group. *Allobaculum*, a genus of SCFA-producing bacteria, has been associated with improvements in obesity, such as reduced body weight and diminished low-grade inflammation, as well as better insulin resistance management. Additionally, a decrease in the *Akkermansia* population is inversely related to adiposity and fat accumulation in animals, increasing with intestinal mucin as the major precursor of carbon and nitrogen. Taken together, we noticed that HFD modulates *Firmicutes* and *Proteobacteria* while altering *Allobaculum* and *Akkermansia* in the HFD group, which might be involved in AT inflammation during obesity.

### 3.3. Specific Phylotype Alteration in the Fecal Microbiome in HFD-Fed Mice

*Lactobacillus,* associated with obesity and metabolic diseases, had a higher abundance in the stool samples collected from obese individuals. On the contrary, this genus was found in a low abundance in the commensal gut microbiota [37]. Further, *Akkermansia* was a highly abundant taxon in the ND group; however, *Lactobacillus* and *Ruminococcus* exhibited greater prominence in the HFD group in this study. To identify the alteration in leading bacteria, we employed linear discriminant analysis (LDA) and effect size (LEfSe) analysis to examine the varying levels of dominant microbiota in each group (Figure 3A,B). Figure 3 depicts a total of 22 different taxa, split into 2 groups, with 11 phylotypes as key biomarkers of notable gut microbiota in each of the ND and HFD groups (Figure 3A). The cladogram showing the gut microbiota of primary bacteria exhibited an obvious disparity in the taxa between the HFD and ND groups (Figure 3B). At the phylum level, *Bacteroidetes* and *Firmicutes* in the ND and HFD groups were notably enriched and distinct, but at the class level, *Bacteroidia*, *Clostridia*, and *Erysipelotrichia* distinctly varied between different groups. At the family level, the abundance of *Muribaculaceae*, also known as S24-7, was significantly diverse in the ND group, whereas *Streptococcaceae*, *Clostridiaceae*, *Lachnospiraceae*, *Ruminococcaceae*, and *Erysipelotrichaceae* were the diverse population in HFD group. At the genus level, the abundance of unclassified *Muribaculaceae* and *Allobaculum* was unique in the ND group, but *Lactococcus* and unclassified *Erysipelotrichaceae* were enhanced populations in the HFD group. The heatmap of Spearman’s correlation analysis between the gut microbiome of ND- and HFD-fed mice also showed a strong correlation between 30 taxa (Figure 3C). Taken together, we noticed a significant increase in *Arcobacter*, *Coprococcus*, *Oscillospira*, and *Lactobacillus* in HFD-fed mice compared to those fed an ND.

### 3.4. HFD Differentially Modulates Bacterial Populations

Next, hierarchical clustering, coupled with a heatmap analysis, was conducted to unveil the unique characteristics of the significantly diverse bacterial species based on the abundance of the identified bacteria in the ND and HFD groups. We noticed that the abundance of *Akkermansia muciniphilla*, *Actinobacter lwoffii*, *Clostridrium perfringens*, and *Rhodoplanes elegans* was downregulated in the HFD groups compared to those fed an ND. On the contrary, HFD feeding upregulated the abundance of *Faecalibacterium prausnitzii*, *Actinobacter rhizosphaerae*, *Bacillus cereus*, *Bacillus selenatarsenatis*, and *Actinobacter johnsonii* compared to ND feeding (Figure 4A). We utilized the detection threshold graph to filter and represent only the bacterial genera to ensure that the focus was on the most significantly changed microbiota (Figure 4B). The prevalence value was set from 0 to 1: the higher the value, the higher the prevalence of the sample. However, setting a threshold might have also excluded some significant taxa having low abundance, which could be subjected to further analysis. In our study, only those genera with relative abundances exceeding the threshold of 0.1% were shown, making it easier to identify the dominant and relevant taxa within the microbial community. Overall, HFD feeding showed a significant depletion of commensal gut bacterial species compared to ND-fed experimental animals, which might be the reason for sustained obesity and AT inflammation after HFD feeding.

### 3.5. Correlation and Functional Enrichment Analysis of Microbiome–Metabolite Interaction

The human gut microbiome plays an important role in fatty acid and lipid metabolism, significantly impacting host health. Particularly, Bacteroidetes and Firmicutes phyla play a crucial role in breaking down complex lipids and synthesizing essential fatty acids [38]. A novel study has reported a decrease in Turicibacter after HFD feeding [39]. Monosaturated fatty acids (MUFAs) are considered beneficial for health, and Tunicibacter is reported to have a positive correlation with MUFAs [40]. These microbes produce metabolites like short-chain fatty acids (SCFAs) and bile acids, which ultimately maintain lipid metabolism and energy homeostasis. To demonstrate the dysregulation of the gut microbiota population, we performed a Taxon Set Enrichment Analysis (TSEA) to demonstrate insights into ecological dynamics or functional relevance. In this study, most of the prominently enriched or depleted taxa were represented, and correlation analysis was performed (Figure 5A). HFD feeding significantly upregulated the *Sporosarcina* population compared to ND-fed mice. Interestingly, despite the significant increase in the Firmicutes phylum with HFD feeding, the abundance of *Turicibacter*, a genus within this phylum, was notably reduced in our study.

The enriched KEGG (Kyoto Encyclopedia of Genes and Genomes) pathways in the microbiome’s metabolic pathway were studied to represent the functional processes significantly over-represented in a microbial community after HFD feeding. Lipid metabolism in the microbiome can influence the host’s lipid profiles, impacting conditions such as cardiovascular disease, obesity, and inflammation [41]. We identified 10 microbiota-perturbed pathways under HFD-fed conditions (Table 1). Our analysis included a significant change in many pathways related to amino acid and lipid metabolism, including the pentose phosphate pathway, phenylalanine, tyrosine, and tryptophan biosynthesis, valine, leucine, and isoleucine biosynthesis, thiamine metabolism, folate biosynthesis, etc. The bipartite network graph below depicts the microbe–metabolite interactions through a functional enrichment analysis (Figure 5B). The graph illustrates the bacterial species contributing to the production or modification of specific metabolites. This can help identify key microbial players in particular metabolic pathways. For instance, “2-Dehydro-3-deoxy-D-arabino-heptonate” and “Deoxyribose 1-phosphate” appear to have significant interactions with multiple bacterial species. Some bacterial species, such as *Bifidobacterium longum* and *Prevotella stercorea*, are connected to different metabolites, indicating their crucial role in microbial metabolism. Many bacterial species are also found to be associated with 2-Methyl-4-amino-5-hydroxymethylpyrimidine diphosphate. Overall, these findings highlight specific bacterial taxa and their associated metabolites that contribute to inflammation, insulin resistance, or energy balance.

## 4. Discussion

The human gut microbiome has become a widely investigated research area and is increasingly recognized as a significant contributor to various host metabolic disorders, such as obesity, type 2 diabetes (T2D), non-alcoholic fatty liver, and cardiovascular disease [42]. Several studies have shown differential changes in alpha- and beta-diversity of the gut microbiota due to HFD feeding [43]. A low alpha-diversity is often indicative of a dysbiotic gut microbiome and is clinically associated with conditions such as obesity [44]. Although a clear definition of a “healthy gut microbiota” remains elusive, numerous disease conditions have been linked to microbiota compositions that differ from those of healthy control groups [45]. Research indicates that the microbiota associated with obesity is more efficient at extracting energy from one’s diet, suggesting its pivotal role in the storage of excessive energy [3]. HFDs have consistently been shown to modulate gut microbial composition in humans [46]. The gut microbes may influence fat metabolism indirectly through intestinal signaling regulated by SCFAs and lactate, such as butyrate, acetate, and propionate, which are produced through the fermentation of carbohydrates and serve both as energy sources and signaling molecules for the host [17]. In this study, we observed that HFD feeding reduced the richness of the gut microbiota population while increasing its diversity and caused a higher abundance of *Firmicutes* compared to ND feeding. Further, the total sum scaling (TSS) analysis revealed a significant decrease in the *Akkermansia* and *Allobaculum* populations in the HFD group compared to the group fed ND. Our linear discriminant analysis also showed a decreased abundance of butyrate-producing *Muribaculaceae* in the HFD group. Taken together, our studies highlight that HFD modulates α-diversity and β-diversity of the gut microbiome and facilitates the growth of obesity-promoting microbiota and, therefore, obesity itself.

As a prominent mucin utilizer, *Akkermansia muciniphila* has gained significant attention as a potential next-generation probiotic and is considered crucial for maintaining the intestinal epithelial barrier [47,48]. A recent clinical trial demonstrated that a 3-month oral administration of *Akkermansia muciniphila* in obese patients is safe and well tolerated. Supplementation with *Akkermansia muciniphila* leads to reductions in body weight and improvements in liver function and inflammation, as evidenced by a double-blind, randomized human study [49]. Our study corroborates this finding, as we noticed a depleted population of *Akkermansia* in the HFD group compared to the ND group. The *Muribaculaceae* family, a prominent member of the *Bacteroides* genus, was found to have a significant negative correlation with the risk of obesity. Regarding *Muribaculaceae,* novel data on the diversity, ecology, and description of the bacterial family S24-7 have been reported [50]. It has been shown that early-life gut microbiome modulation alters gut responses, contributing to accelerated and enhanced T1D development [51].

Our findings demonstrate a significant decrease in Muribaculaceae under HFD feeding, which supports the above finding about its role in obesity and metabolic diseases.

In the past, it was established that changes in the microbiome modulate inflammation [52]. Chronic low-grade inflammation is a hallmark of different metabolic diseases, including obesity, T2D, and non-alcoholic fatty liver disease [53,54]. During obesity, the rapid expansion of adipose tissues leads to hypoxic conditions and releases pro-inflammatory cytokines and chemokines. This subsequently triggers the infiltration of various immune cells, including macrophages and T cells, secreting pro-inflammatory cytokines like tumor necrosis factor (TNF), interleukin-1 beta (IL-1β), and IL-6 [55,56]. We noticed an increased abundance of *Ruminococcus gnavus* in the hierarchical clustering heatmap in our study. This species is correlated with an increased expression of haptoglobin in normal-weight and overweight individuals [57]. Haptoglobin can attract monocytes/macrophages by interacting with chemokine receptor 2 (CCR2), a process mediated by MAPK phosphorylation [58]. Haptoglobin levels rise in white adipose tissues with weight gain, activating macrophages and triggering the release of TNF-α and IL-6 [59]. It has been observed that the MAPK p38 inhibitor reduces the phosphorylation of CCAAT/enhancer-binding protein (C/EBP)-α, peroxisome proliferator-activated receptor-γ (PPAR-γ) and its downstream effector, activating transcription factor-2 (ATF-2) [60]. HFD feeding not only alters gut microbiota composition and induces functional changes in the intestinal barrier but can also facilitate low-grade inflammation in the small intestine by leaking LPS, ultimately causing systemic inflammation [61]. Further, LPS is involved in the transition from M2 to M1 macrophages, the latter of which accelerates inflammatory responses in adipose tissues. Additionally, LPS may activate caspase-4/5/11 signaling, leading to cell death in adipocytes [62]. In mice, the inflammatory caspases-1 and -11 play a critical role as key mediators of septic shock in vivo [63,64]. Caspase-11 has been identified as a pivotal regulator of non-canonical inflammasome activation, accelerating the release of IL-1α in response to intracellular LPS and Gram-negative bacteria. It also plays a critical role in inducing inflammasome-associated cell death [65,66]. Based on amino acid sequences, caspase-4 and caspase-5 are considered the potential human orthologues of murine caspase-11, and previous studies have identified caspase-4 and caspase-5 as critical downstream targets of LPS activation in human monocytes. Further, caspase-5 specifically contributes to the release of IL-1α and IL-1β, but not IL-6, from LPS-stimulated monocytes [67]. The infiltration of macrophages, along with the upregulation of TNF-α, NF-κB, and IL-6, further promotes inflammation in adipose tissue [68,69]. In this study, we found a dramatic increase in the population of several LPS-producing microbiomes at the genus level, including *Klebsiella* and *Pseudomonas*, which are well reported to produce LPS [70,71]. *Lactobacillus* has been claimed to be an obesity-associated taxon, with higher abundance observed in the stool of patients with obesity and metabolic diseases. In our correlation analysis using TESA, we found a remarkable increase in *Lactobacillus*. This probiotic genus, derived from food, exhibits relatively low prevalence and abundance in the commensal gut microbiota [37]. Gut dysbiosis has been linked to several pathological conditions of the gastrointestinal tract, including diarrhea and irritable bowel syndrome, as well as other immune system-related disorders like inflammatory bowel diseases and rheumatoid arthritis [72,73]. Evidence indicates that systemic inflammatory mediators from dysbiotic bacteria impair the blood-brain barrier’s integrity and facilitate neuroinflammation and neurodegeneration, which ultimately lead to the development of Alzheimer’s disease [74]. Several studies have demonstrated a direct and indirect relationship between the gut microbiota population with the development and progression of several cancers. HFD consumption is well reported to promote colorectal cancer (CRC), and multiple studies have shown that altered adipokines, bile acids, and, most importantly, gut microbiota dysbiosis contribute to CRC development [75,76]. Patients with advanced-stage breast cancer often show a greater abundance of *Clostridium* and *Lachnospiraceae* species in their feces compared to those with early-stage breast cancer, pointing to the connection between gut microbiota and the progression of breast cancer [77].

Exploring the correlation between dysregulated gut microbes and metabolites is important for alterations in obesity and HFD-induced diabetes. Our microbe metabolite interaction study identified several potential metabolites connected to several bacterial species. An imbalance in “good” and “bad” gut microbiota resulted in a depletion of bacterial metabolites like SCFAs and activated the HFD–gut microbiota–butyrate–insulin resistance pathway in diabetes induced by HFD [78,79]. In our study, we noticed an increase in *Ruminococcus*, a species that might be responsible for the degradation of polysaccharides and fibers. Further, we found a significant increase in *Firmicutes* at the phylum level in mice following a high-fat diet (HFD), a finding which had been previously demonstrated in other animal and human studies [80]. *Firmicutes* are the main functional population extracting energy from dietary carbohydrates, and this enhanced energy harvest leads to caloric absorption and, ultimately, weight gain [81]. Our enriched KEGG pathway analysis showed 10 significantly enriched metabolic pathways after HFD feeding. Cysteine and methionine metabolism and phenylalanine, tyrosine, and tryptophan biosynthesis were found to be increased in our KEGG pathway analysis. Valine, leucine, and isoleucine are collectively known as branched-chain amino acids, which are essential in metabolic processes but, at higher levels, can cause insulin resistance and obesity [82]. In our study, we found the valine, leucine, and isoleucine biosynthesis pathway to be one of the most significantly enriched metabolic pathways after HFD administration. Considering all these findings, understanding these complex microbial interactions is essential for developing targeted interventions to mitigate obesity and inflammation by aiming at the gut microbiota and their metabolites.

## 5. Conclusions

High-fat diets (HFDs) have been identified as a primary risk factor for several autoimmune and metabolic diseases. This study provides valuable insights into the dysbiosis of gut microbiota and the alterations in metabolites induced by HFDs. An HFD promotes the development of a gut microbiome that enhances energy extraction from the diet, thereby contributing to obesity, AT inflammation, and metabolic diseases. We found a notable dysregulation in major bacterial populations at different taxonomical levels, including increased levels of *Firmicutes* and depleted populations of *Bacteroidetes* in HFD-fed mice. Additionally, *Akkermansia* and *Allobaculum* were also found to be decreased in our study. It is feasible to alleviate the symptoms of obesity and metabolic diseases by utilizing probiotics to modify the composition of the gut microbiota. There is substantial evidence that the gut microbiota’s composition and their metabolites influence the development of obesity and related diseases. Therefore, targeting the composition of the gut microbiota and the metabolites generated by dietary interventions and medications offers potential strategies for treating and preventing metabolic diseases. In the future, the field of microbiota research will offer novel insights into microbiota–host interactions and pave the way for the development of therapeutic approaches for obesity and its associated complications.

## Figures and Tables

**Figure 1 cells-14-00463-f001:**
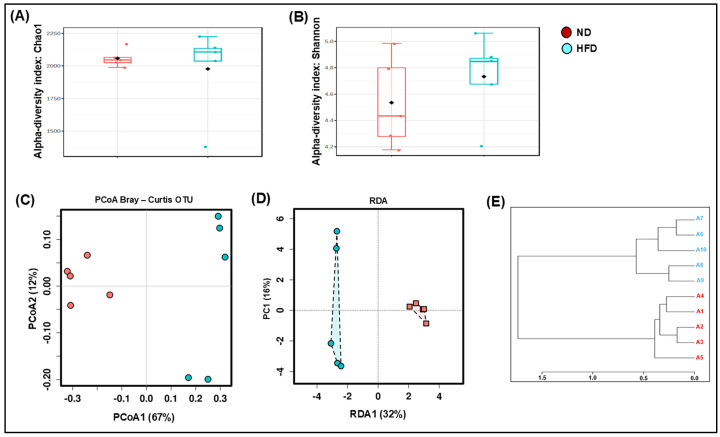
Effect of HFD supplementation on overall presentations of the gut microbiota structure (*n* = 5). Alpha-diversity was estimated with the Chao1 index (**A**) and Shannon index (**B**) on raw OTU abundance in ND- and HFD-fed subjects. Principal coordinate analysis (PCoA) was conducted on bacterial beta-diversity based on the Bray–Curtis dissimilarity of gut microbial populations by relative abundance (**C**), the results were based on the weighted UniFrac distance of bacterial communities from different regions (**D**). This was followed by taxonomic diversity, analyzed by the unweighted paired-group method with an arithmetic mean (UPGMA) (**E**). ND-fed subjects are shown in red, while HFD-fed animals are shown in blue.

**Figure 2 cells-14-00463-f002:**
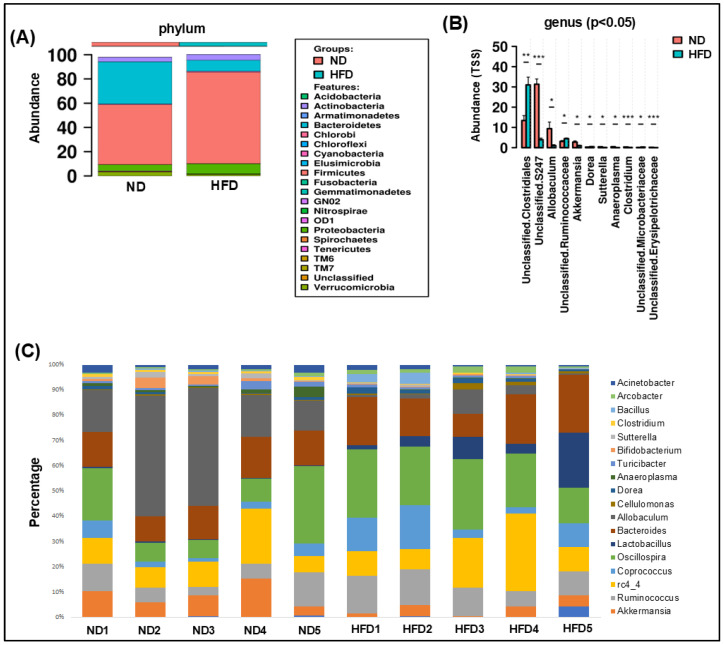
Relative abundance of bacteria at the phyla and genus level (*n* = 5). Relative abundance of community at the phylum level (**A**). Changes in abundance of dominant community at the genus level in ND- and HFD-fed mice (**B**). The relative abundance of bacterial genera in different fecal samples is visualized by bar plots. Each bar represents a subject, and each colored box is a bacterial genus. The height of a color box demonstrates the relative abundance of that organism within the sample (**C**). A *t*-test was performed to assess the significance of the indicated classes. The bars represent the mean ± SEM and exact *p*-values are indicated in the text (* *p* < 0.05; ** *p* < 0.01; and *** *p* < 0.001).

**Figure 3 cells-14-00463-f003:**
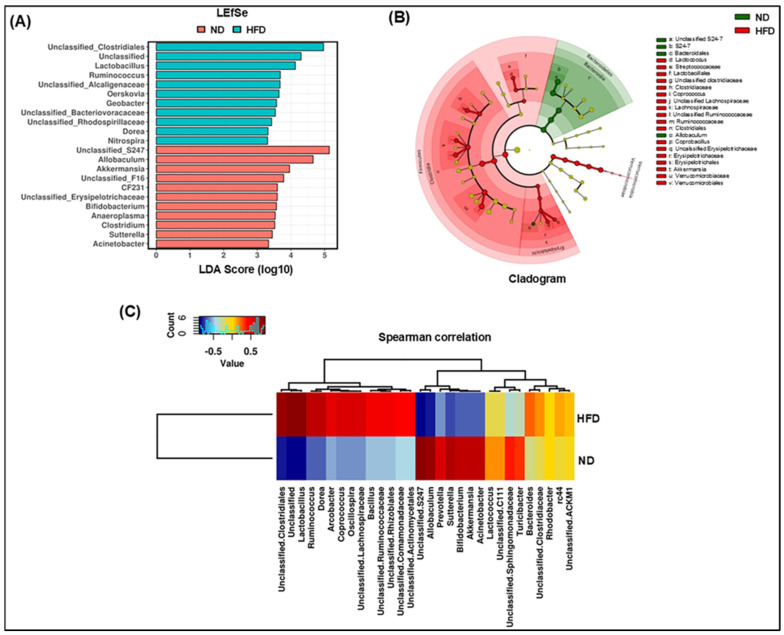
ND and HFD resulted in different gut microbial clusters and different dominant bacteria in the fecal sample. In this study, 16S rRNA sequencing from the fecal samples was performed, and OTU data were subjected to a linear discriminant effect size (LEfSe) analysis (**A**). The cladogram represents the phylogenetic relationship of significant OTUs associated with each group (**B**). The heatmap of Spearman’s correlation analysis of relative abundances in the gut microbiome between ND and HFD groups is also shown above (**C**). For LefSe data, the alpha factorial Kruskal–Wallis test among classes was set to 0.05, and the threshold on the logarithmic LDA score for discriminative features was set to 3.

**Figure 4 cells-14-00463-f004:**
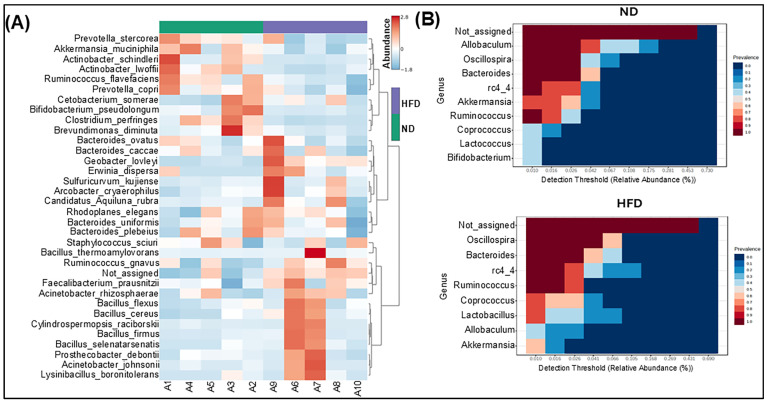
Differentially abundant genera and species in ND and HFD groups: heatmap of the proportion of OTUs determined to be dominant bacteria, with rows clustered by microbiota similarity according to the Euclidean distance, and columns clustered by OTUs occurring more often together (**A**); and core microbiome showing genera detected in high fractions in ND and HFD groups (**B**).

**Figure 5 cells-14-00463-f005:**
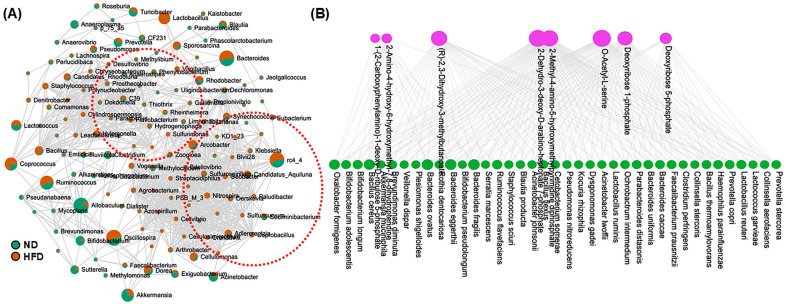
Correlation analysis of gut microbiome between ND and HFD groups and microbiome–metabolite interactions. Gut microbiome correlation network for genus classification is shown in (**A**). The nodes represented the taxa at the genus level according to the relative content between ND and HFD groups. The green color in the nodes represents the ND group, whereas the red color in the nodes represents the HFD group. The edges represent correlations between the taxon pairs. Interaction between gut microbiota at the species level and specific microbiota is shown in (**B**), in a diagram illustrating significant microbiome–metabolite associations. The nodes represent microbial taxa (green) and metabolites (purple), while the edges indicate significant positive correlations.

**Table 1 cells-14-00463-t001:** Enriched KEGG pathways.

Metabolic Pathway (KEGG)	Total	Expected	Hits	*p*-Value	FDR	Features
Pentose phosphate pathway	35	0.0453	2	0.000853	0.156	C00672
Phenylalanine, tyrosine, and tryptophan biosynthesis	35	0.0453	2	0.000853	0.156	C00673
Sulfur relay system	11	0.0142	1	0.0142	1	C01302
Valine, leucine, and isoleucine biosynthesis	23	0.0298	1	0.0294	1	C04272
Pantothenate and CoA biosynthesis	28	0.0362	1	0.0357	1	C04272
Thiamine metabolism	31	0.0401	1	0.0394	1	C04752
Sulfur metabolism	33	0.0427	1	0.0419	1	C00979
Zeatin biosynthesis	39	0.0505	1	0.0494	1	C00979
Folate biosynthesis	57	0.0738	1	0.0715	1	C01300
Cysteine and methionine metabolism	63	0.0815	1	0.0787	1	C00979

KEGG pathways were explored using Gene set enrichment analysis (GSEA) 2.0. Each metabolite pathway shows the *p*-value after the false discovery rate (FDR) and other features. For each pathway, an FDR is analyzed, and the value represents the statistical significance of the enrichment. By using GSEA user guidelines, our experimental condition suggested a threshold of significance FDR #0.05. Due to the exploratory nature of this study, we took a more conservative approach to the threshold of significance (FDR# 0.01). Leading-edge analysis was also used in case of overlapping of various pathways.

## Data Availability

The raw data of this manuscript will be made available to the public by the authors, without any reservation.

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
