# Peer review of "Impact of a High-Fat Diet on the Gut Microbiome: A Comprehensive Study of Microbial and Metabolite Shifts During Obesity"

_cells, 2025, doi:10.3390/cells14060463_

Round 1
Reviewer 1 Report
Comments and Suggestions for Authors
This research is very interesting. It addresses a burning issue and the manuscript is well structured. However, there are some shortcomings that should be addressed.
In the INTRODUCTION section (lines 62-67), harmful bacteria are mentioned, including species from the genera Dorea and Candida. However, Candida is not a bacterium but a fungus (specifically a yeast). This sentence should be expressed differently. So should the following one (speaking of micro-organism or microbiological species, for example).
Line 95 talks about ND but it has not been previously stated what these acronyms correspond to (only in the ABSTRACT has Normal Diet been mentioned). It should be specified what the acronym ND corresponds to.
Two ideas should be included in some section of the DISCUSSION:
i) To emphasise that beta-diversity is related to a healthy state of the organism; therefore, if high-fat diets (HFD) decrease beta-diversity (or promote alpha-diversity, which has similar consequences), it implies that HFD somehow antagonises a healthy state. We think that this idea should be clear to potential readers of this manuscript.
ii) Other microbiota-related conditions (or dysbiosis, for that matter) are mentioned in lines 363-365, but this mention is too brief. Neurodegenerative diseases, cancer and digestive diseases should also be mentioned. We believe this would better highlight the importance of this research, as well as its potential translation to the humans.
Author Response
Reviewer #1 Response:
We thank the reviewers for their insightful comments that contributed to improving the current version, and we are extremely grateful for the opportunity to revise our submission. We carefully considered all suggestions and have provided detailed responses addressing the raised issues. We hope that the revised manuscript, with all modifications, font color in red for your attention.
- 1. In the INTRODUCTION section (lines 62-67), harmful bacteria are mentioned, including species from the genera Doreaand Candida. However, Candidais not a bacterium but a fungus (specifically a yeast). This sentence should be expressed differently. So should the following one (speaking of micro-organisms or microbiological species, for example).
Response:
We thank the reviewer for this profound oversight of the manuscript. We have modified this sentence differently to avoid any discrepancy in the revised version.
2. Line 95 talks about ND but it has not been previously stated what these acronyms correspond to (only in the ABSTRACT has Normal Diet has been mentioned). It should be specified what the acronym ND corresponds to.
Response:
We greatly appreciate this suggestion regarding the full form of ND. We have added a normal diet as the elaboration of (ND in line 93), where it first appeared after the abstract in the revised manuscript.
- 3. Two ideas should be included in some section of the DISCUSSION:
- i) To emphasize that beta-diversity is related to a healthy state of the organism; therefore, if high-fat diets (HFD) decrease beta-diversity (or promote alpha-diversity, which has similar consequences), it implies that HFD somehow antagonizes a healthy state. We think that this idea should be clear to potential readers of this manuscript.
- ii) other microbiota-related conditions (or dysbiosis, for that matter) are mentioned in lines 363-365, but this mention is too brief. Neurodegenerative diseases, cancer, and digestive diseases should also be mentioned. We believe this would better highlight the importance of this research, as well as its potential translation to humans.
Response: These are excellent points raised by the reviewer. We greatly appreciate your constructive suggestions, which will help to make this manuscript more meaningful. We have modified our discussion by emphasizing the modulation of alpha and beta diversity through a high-fat diet (HFD) (lines 437- 445). We have also addressed the context of other diseases like neurodegenerative diseases, cancer, and digestive diseases in the relevant section of the discussion (lines 544-556).
Reviewer 2 Report
Comments and Suggestions for Authors
This manuscript investigates how a high-fat diet (HFD) influences the gut microbiome and its metabolites in mice. Using 16S rRNA sequencing and metabolomic analyses, the study reports an increase in Firmicutes and a decrease in Bacteroidetes, along with a reduction in beneficial genera such as Akkermansia and Allobaculum. These microbial shifts are associated with enhanced energy extraction and inflammation. The findings underscore the gut microbiome’s role in diet-induced metabolic dysregulation and suggest microbiota-targeted interventions as potential therapeutic strategies for obesity-related disorders. Overall, this manuscript provides valuable insights into HFD-induced dysbiosis but requires revisions to improve clarity, reproducibility, and accuracy.
1. Line 73: The statement, "SCFAs have been shown to activate the carbohydrate-responsive element-binding protein (ChREBP) and the sterol regulatory element-binding transcription factor 1 (SREBP1), both of which are involved in lipogenesis," requires further support. While SCFAs can influence lipogenesis, their primary role is as energy substrates or signaling molecules via G-protein-coupled receptors (e.g., GPR41/43). Direct activation of ChREBP and SREBP1 is less well-established and typically linked to glucose or insulin signaling. The supporting citations should be provided.
2. Line 273: The font in Section 3.5 is inconsistent with the rest of the manuscript and should be standardized.
3. Line 293: The claim, "We have identified 10 microbiota-perturbed pathways under HFD-fed conditions (Table 1)," lacks sufficient statistical justification, as most pathways have only a single hit, and the false discovery rate (FDR) is reported as 1. Further details on statistical thresholds (e.g., what constitutes significance in KEGG analysis) should be provided to ensure the conclusions align with reported hits and FDR values.
4. Line 372: The activation of "caspase-4/5/11" by LPS is mentioned without specifying the species context. Since mice possess caspase-11, while humans have caspase-4/5, this distinction should be clarified.
5. The authors conclude that gut microbiome and metabolite shifts "regulate" adipose tissue inflammation and obesity. However, the results only describe changes in gut microbiota and metabolite profiles following HFD feeding without direct evidence of a regulation evidence. The title should be revised to reflect the study’s observational nature.
6. The manuscript frequently states that a high Firmicutes-to-Bacteroidetes ratio is linked to obesity. However, recent studies challenge this assumption, with some reporting inconsistent or opposite findings (e.g., 10.1371/journal.pone.0255446, 10.3345/cep.2021.01837). It would be beneficial to acknowledge this controversy and discuss potential confounding factors influencing the F/B ratio.
Author Response
Reviewer # 2 Response:
- Line 73:The statement, "SCFAs have been shown to activate the carbohydrate-responsive element-binding protein (ChREBP) and the sterol regulatory element-binding transcription factor 1 (SREBP1), both of which are involved in lipogenesis,"requires further support. While SCFAs can influence lipogenesis, their primary role is as energy substrates or signaling molecules via G-protein-coupled receptors (e.g., GPR41/43). Direct activation of ChREBP and SREBP1 is less well-established and typically linked to glucose or insulin signaling. The supporting citations should be provided.
Response:
We would like to thank the reviewer for these meaningful suggestions. We acknowledge that direct activation of ChREBP and SREBP1 is not well-established. Due to this reason, we have deleted this sentence and added a new sentence to avoid confusion and improve readers' understanding of this manuscript.
- Line 273:The font in Section 3.5 is inconsistent with the rest of the manuscript and should be standardized.
Response:
Thank you for pointing out this mistake from our end. We have carefully reviewed the formatting in Section 3.5 and have standardized the font to ensure consistency throughout the manuscript. We appreciate your attention to detail and have made the necessary corrections to address this issue.
- Line 293:The claim, "We have identified 10 microbiota-perturbed pathways under HFD-fed conditions (Table 1),"lacks sufficient statistical justification, as most pathways have only a single hit, and the false discovery rate (FDR) is reported as 1. Further details on statistical thresholds (e.g., what constitutes significance in KEGG analysis) should be provided to ensure the conclusions align with reported hits and FDR values.
Response:
KEGG pathways were explored using Gene set enrichment analysis (GSEA) 2.0. Each metabolite pathway shows the p-value after the False Discovery Rate (FDR) and features. For each pathway, an FDR is analyzed, and the value represents the statistical significance of the enrichment. By using GSEA users’ guidelines our experimental condition suggests a threshold of significance FDR #0.05. Due to the exploratory nature of this study, we took a more conservative approach for a threshold of significance (FDR# 0.01) The Leading-edge analysis was also used in case of overlapping among various pathways.) This has been added in the revised manuscript line number 432-437.
- Line 372:The activation of "caspase-4/5/11" by LPS is mentioned without specifying the species context. Since mice possess caspase-11, while humans have caspase-4/5, this distinction should be clarified.
Response:
Thank you for highlighting this important point. We have modified the manuscript to specify the species context when discussing the activation of caspase-4/5/11 by LPS. In our revised version of the manuscript, we clearly stated that mice possess caspase-11, while humans have caspase-4 and caspase-5 (Lines 490 to 498). This clarification will avoid any potential confusion for readers. We appreciate your valuable feedback on this matter.
- The authors conclude that gut microbiome and metabolite shifts "regulate"adipose tissue inflammation and obesity. However, the results only describe changes in gut microbiota and metabolite profiles following HFD feeding without direct evidence of regulation evidence. The title should be revised to reflect the study’s observational nature.
Response:
Thank you for your valuable input. We appreciate the importance of precise language to reflect the observational nature of our study. We carefully reconsidered the title and modified it in our revised version of the manuscript.
- The manuscript frequently states that a high Firmicutes-to-Bacteroidetes ratio is linked to obesity. However, recent studies challenge this assumption, with some reporting inconsistent or opposite findings (e.g., 10.1371/journal.pone.0255446, 10.3345/cep.2021.01837). It would be beneficial to acknowledge this controversy and discuss potential confounding factors influencing the F/B ratio.
Response:
Thank reviewers for raising this important oversight. We acknowledge that there is ongoing debate in the field regarding the association between the Firmicutes-to-Bacteroidetes (F/B) ratio and obesity. We intended to highlight the F/B ratio's potential as a marker within the broader context of gut microbiota composition and its relationship to obesity. However, we agree that acknowledging this controversy will strengthen the manuscript’s scientific rigor and objectivity. Therefore, we have revised the relevant section of the manuscript.
Round 2
Reviewer 2 Report
Comments and Suggestions for Authors
The authors addressed all my questions.